# Effect of the Consumption of Alcohol-Free Beers with Different Carbohydrate Composition on Postprandial Metabolic Response

**DOI:** 10.3390/nu14051046

**Published:** 2022-02-28

**Authors:** Itziar Lamiquiz-Moneo, Sofia Pérez-Calahorra, Irene Gracia-Rubio, Alberto Cebollada, Ana M. Bea, Antonio Fumanal, Ana Ferrer-Mairal, Ascensión Prieto-Martín, María Luisa Sanz-Fernández, Ana Cenarro, Fernando Civeira, Rocio Mateo-Gallego

**Affiliations:** 1Laboratorio de Investigación Molecular, Hospital Universitario Miguel Servet, Instituto de Investigación Sanitaria Aragón (IIS Aragón), Centro de Investigación Biomédica en Red Enfermedades Cardiovasculares (CIBERCV), Universidad de Zaragoza, 50009 Zaragoza, Spain; irenegraciarubio@gmail.com (I.G.-R.); anamariabeasanz@hotmail.com (A.M.B.); ana.cenarro@gmail.com (A.C.); civeira@unizar.es (F.C.); rmateo@unizar.es (R.M.-G.); 2Departamento de Anatomía e Histologías Humanas, Facultad de Medicina, Universidad de Zaragoza, 50009 Zaragoza, Spain; 3Departamento de Fisiatría y Enfermería, Facultad de Ciencias de la Salud y del Deporte, Universidad de Zaragoza, 22002 Huesca, Spain; s.perez.84@hotmail.com; 4Unidad de Biocomputación, Instituto Aragonés de Ciencias de la Salud (IACS Aragón), 50009 Zaragoza, Spain; acebolladaso.iacs@aragon.es; 5Grupo Ágora—La Zaragozana S.A., 50007 Zaragoza, Spain; antonio.fumanal@agoragp.com (A.F.); aprieto@agoragp.com (A.P.-M.); 6Instituto Agroalimentario de Aragón (IA2), 50013 Zaragoza, Spain; ferrerma@unizar.es; 7Departamento de Producción Animal y Ciencia de los Alimentos, Facultad de Ciencias de la Salud y del Deporte, Universidad de Zaragoza, 22002 Huesca, Spain; 8Servicio de Urgencias, Hospital Clínico Universitario Lozano Blesa, 50009 Zaragoza, Spain; msanzf86@gmail.com; 9Instituto Aragonés de Ciencias de la Salud (IACS), 50009 Zaragoza, Spain; 10Departamento de Medicina, Psiquiatría y Dermatología, Facultad de Medicina, Universidad de Zaragoza, 50009 Zaragoza, Spain

**Keywords:** alcohol-free beer, postprandial effect, glucose metabolism, insulin, incretin hormones

## Abstract

Background: We investigated the postprandial effects of an alcohol-free beer with modified carbohydrate (CH) composition compared to regular alcohol-free beer. Methods: Two randomized crossover studies were conducted. In the first study, 10 healthy volunteers received 25 g of CH in four different periods, coming from regular alcohol-free beer (RB), alcohol-free beer enriched with isomaltulose and a resistant maltodextrin (IMB), alcohol-free beer enriched with resistant maltodextrin (MB), and a glucose-based beverage. In the second study, 20 healthy volunteers were provided with 50 g of CH from white bread (WB) plus water, or with 14.3 g of CH coming from RB, IMB, MB, and extra WB. Blood was sampled after ingestion every 15 min for 2 h. Glucose, insulin, incretin hormones, TG, and NEFAs were determined in all samples. Results: The increase in glucose, insulin, and incretin hormones after the consumption of IMB and MB was significantly lower than after RB. The consumption of WB with IMB and MB showed significantly less increase in glucose levels than WB with water or WB with RB. Conclusions: The consumption of an alcohol-free beer with modified CH composition led to a better postprandial response compared to a conventional alcohol-free beer.

## 1. Introduction

Carbohydrates (CH) are the main source of metabolic energy in the body, with their quality having an essential impact on several health problems [1,2,3]. A high consumption of poor-quality CH can lead to the development of obesity, insulin resistance, or type 2 diabetes mellitus (T2DM), among others [1,4,5,6]. The quality of CH is determined by the amount of fiber, percentage of whole grain, processing, and glycemic responses after absorption [7]. Many studies have demonstrated that high-glycemic-index (GI) diets are associated with a high risk of T2DM, cardiovascular disease, and death [8,9].

Alcohol-free beer is a good source of B-complex vitamins (especially folate) and bioactive components, such as flavonoids and phenolic acids, and contains trace amounts of minerals such as potassium, silicon, and magnesium [10]. The consumption of one regular beer supposes the ingestion of 11.8 g of CH, containing approximately 140 Kcal, which can represent 7% of the total calorie consumption of a standard diet [10]. Although an alcohol-free beer has a lower percentage of CH, this beverage has a relatively high GI, estimated at 80, because of its content in starch and oligosaccharides, such as maltose or maltotriose. Therefore, despite their valuable nutritional content, their consumption should be moderated—especially in those subjects with glucose intolerance or T2DM. Our group recently demonstrated that the intake of an alcohol-free beer (including the substitution of regular CH with low doses of isomaltulose and the addition of a resistant maltodextrin) with meals led to an improvement in insulin resistance in subjects with T2DM and overweight or obesity [11]. Isomaltulose is a disaccharide composed of α-1,6–linked glucose and fructose, which is widely used as an alternative sugar since it leads to a delayed digestion and absorption [12]. Resistant dextrin is a glucose polymer (rich in α-1,4 and α-1,6 linkages) derived from wheat or maize, and is principally fermented in the colon [13]. Both have previously demonstrated beneficial effects in glycemic control and insulin resistance when provided at relatively high doses [12,14]. We hypothesized that the synergic effect of these components within alcohol-free beer could improve postprandial glycemia after main meals, which could have a beneficial impact on glucose metabolism in the mid-term. That beneficial impact could be produced by the direct effect on incretin hormones, such as GIP and GLP-1, which are produced in the pancreas [15]. The consumption of modified-CH beers could produce lower peaks of these incretin hormones, which would generate the lowest peaks of glucose and insulin, and would allow the mechanism of action of these modified-CH beers to be identified. Other studies have demonstrated that consuming CH-containing foods concurrently with other nutrients (such as protein) could improve the postprandial effect of that meal [16]. Thus, our objective was to investigate the acute postprandial effects (including glycemic and lipid metabolism, as well as satiety response) produced after fasting and consumption—with and without combination with simple CH in the form of white bread—of alcohol-free beers in which the CH composition has been modified (by replacing conventional CH in beer with isomaltulose and a resistant dextrin), compared to regular alcohol-free beer, in healthy participants.

## 2. Materials and Methods

### 2.1. Subjects

Eligible volunteers were healthy women and men aged 18–80, with a body mass index (BMI) between 18 and 27.5 kg/m^2^, and steady weight (≤5% of body weight) in the previous six months. Exclusion criteria included (a) the presence of cardiometabolic disease, such as dyslipidemia (total cholesterol > 200 mg/dL, triglycerides > 150 mg/dL, or treatment with lipid-lowering drugs), prediabetes or T2DM (glucose ≥ 100 mg/dL and/or HbA1c ≥ 5.7% or hypoglycemic drugs), or hypertension; (b) the presence of other chronic diseases, including cardiovascular disease, kidney disease (glomerular filtration rate < 45 mL/min), active liver disease, uncontrolled hypothyroidism, and any other condition that could limit compliance with the study; (c) gluten intolerance and/or high alcohol intake (>30 mL/day) on a regular basis; (d) sterols or omega-3 fatty acid supplements; and (e) current treatment with weight loss medications. Volunteers who were eligible according to the inclusion and exclusion criteria were scheduled for a pre-screening visit. Informed consent was obtained at the screening visit along with anthropometric, clinical, and biochemical parameters, in order to confirm eligibility, before randomization.

The study protocol was approved by the local ethical committee institution (Comité de Ética e Investigación Clínica de Aragón, PI19/485). All procedures were in accordance with the ethical standards of that committee.

### 2.2. Study Design

#### 2.2.1. Study 1

Study 1 was a randomized, controlled, double-blind trial with a crossover design including four interventions, with 1-week intervals between them (Figure 1): (a) intake of a drink containing 25 g of glucose; (b) intake of 25 g of CH from 447 mL of regular alcohol-free beer (RB); (c) intake of 25 g of CH from 581 mL of alcohol-free beer with almost complete fermentation of regular CH, enriched with isomaltulose (2.5 g/100 mL) and a resistant maltodextrin (0.8 g/100 mL) (IMB); (d) intake of 25 g of CH from 781 mL of alcohol-free beer with almost complete fermentation of regular CH, enriched with resistant maltodextrin (2.0 g/100 mL) (MB). Participants received a different volume of each beer in order to maintain the isoglucidic amounts, and they consumed the beverage under fasting conditions. The complete nutritional composition of the alcohol-free beers is provided in Appendix A.

#### 2.2.2. Study 2

Study 2 was a randomized, controlled, double-blind trial with a crossover design, lasting 5 weeks, with the following interventions (Figure 1): (a) consumption of 50 g of CH from white bread and 330 mL of mineral water; (b) consumption of 50 g of CH from white bread and 14.3 g of CH from RB (254 mL); (c) consumption of 50 g of CH from white bread and 14.3 g of CH from MB (443 mL); (d) consumption of 50 g of CH from white bread and 14.3 g of CH from IMB (330 mL); (e) consumption of 64.3 g of CH from white bread and 330 mL of mineral water.

As in Study 1, participants received a different volume of each beer in order to maintain the isoglucidic amounts, and they consumed the beverages and the bread under fasting conditions. The order of the interventions corresponding to weeks 2, 3, 4, and 5 was variable for each subject. In both studies, the participants were randomized to one or another sequence of intervention using a computerized system (Random.org, accessed on 16 December 2019). Volunteers and research staff were blinded to the type of beer that individuals were assigned to consume in each phase in both studies. Only the researcher who prepared the beverages was aware of the type of beers, but she was not involved in biochemical or statistical analysis of the results. A local brewery prepared the beers in similar containers so as to maintain the blinding, which was only revealed after the results were analyzed.

### 2.3. Diet and Physical Activity

Eligible participants were urged to maintain their lifestyle as stably as possible throughout both studies. Diet and physical activity were assessed across the study to monitor lifestyle factors that could interfere with the studies’ findings. Participants were asked to complete a 24 h weighed food record for 3 days before each visit. Dietary analysis was performed by EasyDiet^®^ (Biocentury, S.L.U, Barcelona, Spain), which is based on Spanish food composition tables [17]. The International Physical Activity Questionnaire (IPAQ)—a brief validated exercise questionnaire—was completed by direct interview with participants [18]. Physical activity was assessed on each visit in both studies.

### 2.4. Anthropometric and Clinical Parameters

Clinical and anthropometric parameters were collected at the beginning of each visit in both studies. Body weight was measured in subjects without shoes to the nearest 0.1 kg using a calibrated scale (SECA 813, Hamburg, Deutschland). Height was assessed to the nearest 0.1 cm with a wall-mounted stadiometer (SECA 2017, Hamburg, Deutschland). BMI was calculated as weight in kilograms divided by the square of height in meters. Waist circumference was measured with anthropometric tape midway between the lowest rib and the iliac crest. Blood pressure was determined at the beginning and at the end of each visit in both studies using a validated semiautomatic oscillometer (Omron HEM-907-E, Hoofddorp, The Netherlands). All measurements were taken in accordance with the recommended guidelines—no food or drink for 3 h prior to measurements, no exhausting exercise for 12 h prior to measurements, and no alcohol or caffeine consumption for 24 h prior to measurements.

### 2.5. Laboratory Measurements

Blood samples were drawn by venipuncture after 10–12 h of fasting in the pre-screening visit, and biochemical parameters for the assessment of eligibility criteria were analyzed. Levels of total cholesterol, triglycerides (TG), HDL cholesterol, gamma glutamyl transferase (GGT), glutamic–pyruvic transaminase (GPT), and glutamic oxaloacetic transaminase (GOT) were measured using standard enzymatic methods. LDL cholesterol levels were calculated with Friedewald’s formula. Non-HDL cholesterol was calculated as total cholesterol minus HDL cholesterol. Blood glucose concentration was measured using the glucose oxidase method. Insulin levels were measured by radioimmunoassay. HbA1c was determined via high-performance liquid chromatography. Thyroid-stimulating hormone (TSH) was determined by TSH3-Dxi luminescent immunoassay (Beckman, Indianapolis, IN, USA).

In each study intervention, blood samples were collected via peripheral venous catheter at baseline (fasting status) and every 15 min for 120 min, after consumption of the drink and the bread, if applicable. Glucose, insulin, gastric inhibitory polypeptide (GIP), glucagon-like peptide 1 (GLP-1), TG, and non-esterified fatty acids (NEFAs) were determined every 15 min in both studies. HDL and LDL cholesterol, leptin, ghrelin, peptide Y (PY), pancreatic polypeptide (PP), and glucagon were determined every 60 min (baseline, 60, and 120 min) in Study 2.

GIP, GLP-1, leptin, ghrelin, glucagon, peptide Y (PY), and pancreatic polypeptide (PP) were determined in plasma using the Human Metabolic Hormone Magnetic Bead Panel protocols from the MILLIPLEX^®^ MAP Kits (Cat. # HMHEMAG-34K, Merck (Darmstadt, Germany)), according to the manufacturer’s instructions. Plasma sample dilutions were carried out according to the detection range of each panel. Assay sensitivities were 0.6 pg/mL for GIP, 2.5 pg/mL for GLP-1, 13 pg/mL for glucagon, 41 pg/mL for leptin, 2 pg/mL for PP, and 28 pg/mL for PYY. Intra-assay precision (mean of % CV) was <10% for all metabolites, while inter-assay precision (mean of % CV) was <15% for all of them. Accuracy was 103% recovery in plasma samples for GIP and GLP-1, 101% recovery in plasma samples for glucagon, 102% recovery in plasma samples for leptin, 104% recovery in plasma samples for PP, and 107% recovery in plasma samples for PY.

NEFAs were determined via colorimetric enzymatic assay using the MaxDiscoveryTM NEFA assay kit from Bioo Scientific Corporation (Catalog # 5620-01), according to the manufacturer’s instructions. Briefly, we added 4 μL of each sample or standard, in duplicate, to the microplate wells, plus 225 μL of NEFA reagent A and 75 μL of NEFA reagent B to each well. We then allowed the plate to stand for 10 min at room temperature, and we measured the absorbance of each well at 550 nm.

### 2.6. White Bread Analysis

Available carbohydrates in the white bread used in Study 2 were analyzed with the available carbohydrate and dietary fiber kit (K-ACHDF 06/18, Megazyme, Bray, Ireland). Briefly, available carbohydrates were determined on triplicate samples of dried ground bread. Samples were incubated at 80 °C with heat-stable α-amylase and incubated at 60 °C with protease and amyloglucosidase. An aliquot was removed, centrifuged, and incubated with a mixture of sucrase/maltase plus β-galactosidase. D-glucose and D-fructose were determined spectrophotometrically using hexokinase plus glucose-6-phostate dehydrogenase, followed by phosphoglucose isomerase. Results are given as available carbohydrates (g/100 g) on an “as is” basis.

### 2.7. Statistical Analysis

Glucose variation after consumption of each beer was established as the main outcome, and its variability in healthy subjects was estimated at 18.4 mg/dL. We expected a difference in glucose variation of 19.4 mg/dL after consumption of RB versus MB. A total sample size of 20 subjects in Study 2 was obtained by considering 90% power (Zβ unilateral = 0.842) to detect a difference between treatment groups and a confidence interval (1-α) of 95% (Zα unilateral = 1.645). Continuous variables are expressed as the mean ± SD or median (25th percentile–75th percentile), as applicable, while categorical (nominal) variables are reported as percentages of the total sample. Differences between independent variables were calculated by *t*-test or Mann–Whitney test, as appropriate, while categorical variables were compared using the chi-squared test. Two-tailed *t*-tests or the Wilcoxon rank sum test for paired samples, as appropriate, were used to compare changes in outcome variables in response to the consumption of each alcohol-free beer. Variations in glucose, insulin, GIP, and GLP-1 over 120 min, after the ingestion of each drink and white bread, if applicable, were analyzed in both studies using ANOVA or the Kruskal–Wallis tests, as appropriate. Differences in the postprandial effects of each intervention were assessed by comparing the area under the concentration versus time curve (AUC) of each metabolite, after the consumption of each drink and white bread, if possible. The AUC and the comparison between them were calculated with the PK package, using the t method. The incremental AUC (iAUC) was calculated with the use of the geometric sums of the areas of the triangles and trapezoids above the fasting glucose concentration over a 2 h period, as previously described [19]. The mixed-design ANOVA model (PROC MIXED) was used to test the differences in glucose, insulin, GIP, and GLP-1 iAUC after the consumption of type of drink in each individual study. All statistical analyses were performed with R version 3.5.0 (R Core Team, Vienna, 2018), and significance was set at *p* < 0.05.

## 3. Results

### 3.1. Participants

A total of 31 participants were examined for study eligibility, of whom 30 met all inclusion/exclusion criteria. Participants were mostly young-adults (31.5 (30.3–34.5) and 30.5 (24.8–33.0) years in Studies 1 and 2, respectively,) with a median BMI of 23.4 (22.4–25.7) and 24.4 (20.6–26.0) kg/m^2^, in study 1 and 2, respectively, and there were more men than women in both studies. All subjects were healthy, with normal values of glucose, TG, and lipid profile, and without any important previous disease—such as cardiovascular disease, hypertension, or T2DM—according to the inclusion and exclusion criteria selection (Table 1). Body weight, blood pressure, physical activity, and dietary characteristics remained stable, without significant differences throughout the two studies. The length of consumption of MB was significantly higher than that of IMB, RB, and glucose in Study 1 (Appendix A). This was an expected finding, since MB had the higher volume (781 mL) in comparison with RB (447 mL) and IMB (581 mL).

### 3.2. White Bread Composition

The white bread analysis reported that the bread used in Study 2 had 53.45 ± 1.73 g of available carbohydrate for each 100 g of total bread.

### 3.3. Glucose Metabolism Parameters

#### 3.3.1. Study 1

The AUC of blood glucose was significantly lower after the consumption of MB than that produced after consumption of RB (*p* = 0.016, Appendix A and Figure 2A1). Moreover, the iAUC of glucose produced after the consumption of MB was significantly lower than that produced after the consumption of glucose or RB (*p* = 0.030 and *p* = 0.024, respectively, Figure 2(A2)). Similarly, the iAUC of glucose produced after the consumption of IMB was significantly lower than that produced after the consumption of glucose or RB (*p* = 0.039 and *p* = 0.032, respectively). In addition, the increase in glucose, calculated as the difference between the basal value and the maximum peak obtained after the ingestion of each drink, was significantly different depending on the drink ingested (*p* = 0.005, Table 2), showing that the higher peak of glucose was produced after the consumption of the glucose-based beverage or RB (Table 2 and Figure 2A).

The AUCs of insulin after the consumption of MB and IMB were significantly lower than those produced after the consumption of RB or the glucose-based beverage (*p* = 0.036, *p* = 0.024, *p* < 0.001, and *p* < 0.001, respectively (Appendix A and Figure 2B1). Moreover, the iAUC of insulin produced after the consumption of MB was significantly lower than that produced after the consumption of glucose or RB (*p* = 0.001 and *p* = 0.001, respectively, Figure 2(B2)). Similarly, the iAUC of insulin produced after the consumption of IMB was significantly lower than that produced after the consumption of glucose or RB (*p* = 0.002 and *p* = 0.001, respectively, Figure 2(B2)). On the other hand, the increase in insulin was significantly different depending on the type of drink that was consumed (*p* = 0.012, Table 2), indicating that the higher increase in insulin levels occurred after the consumption of RB and the glucose-based beverage (Table 2 and Figure 2B).

The AUCs of GIP produced after the consumption of MB and IMB were remarkably smaller than those produced after the consumption of RB and the glucose-based beverage (*p* < 0.001 in all comparisons, Appendix A and Figure 3(A1)). The iAUC of GIP produced after the consumption of MB and IMB was significantly lower than that produced after the consumption of glucose or RB (*p* = 0.011, *p* < 0.001, *p* = 0.009, *p* < 0.001, respectively, Figure 3(A2)). Similarly, increases in GIP were significantly lower after the ingestion of MB or IMB than after the consumption of the glucose-based beverage or RB (*p* < 0.001, Table 2 and Figure 3A).

Regarding to the AUCs of GLP-1, the consumption of the glucose-based beverage produced a significantly lower AUC than those produced after the consumption of the three alcohol-free beers (*p* < 0.001, Appendix A and Figure 3(B1)). However, the iAUC of GLP-1 produced after the consumption of each type of drink did not show significant differences (Figure 3(B2)). Similarly, the increases in GLP-1 did not significantly differ between the alcohol-free beers (Table 2).

#### 3.3.2. Study 2

The lowest AUC of glucose concentration was produced after the consumption of MB with white bread. In fact, the consumption of 64.3 g of carbohydrates, coming from MB plus white bread, generated a significantly lower AUC of glucose than that produced after the consumption of the same amount of CH all sourced from white bread (*p* = 0.012, Appendix A, Figure 4(A1)). Moreover, the AUCs of glucose after the consumption of MB or IMB with white bread were significantly lower than that produced after consumption of RB plus white bread (*p* = 0.006 and *p* < 0.001, respectively, Appendix A and Figure 4(A1)). However, although the lower iAUC was produced after the ingestion of white bread combined with MB, the iAUC produced after the consumption of each type of drink combined with white bread did not show significant differences (Figure 4(A2)). On the other hand, the increase in glucose after each meal intake was significantly different depending on the drink that was consumed (*p* = 0.002, Table 2), showing that the highest increase in glucose concentration was produced after the consumption of RB with white bread (Table 2 and Figure 4A).

The AUC of insulin produced after the consumption of RB plus white bread was significantly higher than that produced after the ingestion of the same amount of CH sourced entirely from white bread (*p* = 0.043, Appendix A and Figure 4(B1)). Similarly, the iAUC produced after the ingestion of white bread combined with RB was significantly higher than the iAUC produced after the consumption of 50 g of white bread (*p* = 0.001, Figure 4(B2)). However, the consumption of MB or IMB with white bread did not lead to significant differences in the AUCs of insulin concentration compared to that produced after the ingestion of the same amount of CH sourced entirely from white bread (*p* = 0.128 and *p* = 0.554, respectively, Appendix A and Figure 4(B1)). Moreover, the consumption of MB plus white bread produced a significantly lower AUC of insulin than that produced after the ingestion of RB combined with white bread (*p* = 0.028, Appendix A and Figure 4(B1)). The increase in insulin was not significantly different depending on the ingested drink (*p* = 0.124, Table 2). Nevertheless, the highest increase in insulin was produced after the consumption of RB combined with white bread (Figure 4B).

The significantly lower AUC of GIP was observed after the ingestion of water with 50 g of CH from white bread. However, with the same amount of CH, only the RB combined with white bread produced a significantly higher AUC of GIP than that produced after the consumption of white bread, without significant differences in the comparison with IMB or MB plus white bread (*p* = 0.019, *p* = 0.857 and *p* = 0.540, respectively; Appendix A, Figure 5(A1)). The highest AUC of GIP was produced after the ingestion of RB combined with white bread, generating a significantly higher iAUC of GIP than that produced after the consumption of 50 g of CH from white bread (*p* = 0.039, Figure 5(A2)). Moreover, the AUC of GIP produced after the consumption of IMB with white bread was significantly lower than that produced after the ingestion of RB plus white bread (*p* = 0.046, Appendix A and Figure 5(A1)). The increase in GIP was significantly different depending on the drink ingested (*p* = 0.011, Table 2).

The lowest AUC of GLP-1 was observed after the ingestion of water with 50 g of CH from white bread (Appendix A and Figure 5(B1)). The lowest iAUC was produced after the ingestion of IMB combined with white bread, which did not show a significant difference from the highest iAUC produced after the consumption of 64.3 g of CH from white bread (*p* = 0.054, Figure 5(B2)). Among alcohol-free beers, significant differences were observed only between the consumption of IMB and white bread, which produced a significantly lower AUC of GLP-1 than that produced after the intake of RB with white bread (*p* = 0.028, Appendix A and Figure 5B). However, there was no significant difference in the increase in GLP-1 depending on the ingested drink (*p* = 0.058) (Table 2).

### 3.4. Hormones Related to Appetite and Satiety

Glucagon, leptin, ghrelin, PY, and PP were analyzed only in the second study at baseline, 60 min, and 120 min. These metabolites did not show significant differences after the consumption of each drink combined with white bread (*p* = 0.988, *p* = 0.723, *p* = 0.924, *p* = 0.992, and *p* = 0.982, respectively). Interestingly, there were no significant differences in these hormones related to appetite and satiety, despite the different CH consumption (Table 3 and Figure 6).

### 3.5. Hormones Related to Appetite and Satiety

#### 3.5.1. Study 1

The AUC and the increase in TG concentration after the intake of different alcohol-free beers and the glucose-based beverage did not significantly differ (*p* = 0.436, Appendix A). Similarly, NEFA concentrations did not show significant differences after the ingestion of each drink, showing similar AUCs and increases from baseline (Appendix A).

#### 3.5.2. Study 2

Total cholesterol, HDL cholesterol, LDL cholesterol, TG, and NEFA concentrations did not significantly change after the consumption of different drinks with white bread. Furthermore, the increase in TG did not significantly differ after the intake of each drink combined with white bread (*p* = 0.812, Appendix A).

## 4. Discussion

The main findings of Study 1 include the following: (a) the consumption of an alcohol free-beer in which regular CH had been almost completely fermented and enriched with isomaltulose and a resistant maltodextrin induced lower glucose and insulin peaks than a regular alcohol-free beer; (b) this effect was analogous after consumption of a similar alcohol-free beer containing a double dose of the resistant maltodextrin but no isomaltulose; (c) these effects were also observed in GIP variation. Study 2’s findings showed the following: (a) consuming 50 g of CH from white bread and an alcohol-free beer in which regular CH had been almost completely fermented, and enriched with isomaltulose and a resistant maltodextrin, induced the same glucose and insulin peaks as the intake of 50 g of CH from white bread plus water; (b) this effect was analogous after the consumption of a similar alcohol-free beer containing a double dose of the resistant maltodextrin but no isomaltulose, along with white bread; (c) both modified alcohol-free beers produced lower glucose, insulin, and GIP increases than the intake of regular alcohol-free beer. Thus, the consumption of an alcohol-free beer—including regular CH fermentation, and adding isomaltulose and a resistant maltodextrin—can effectively impact and improve the postprandial effect of a meal. This effect would be one of the main mechanisms responsible for the findings of a previous study in which the intake of this alcohol-free beer with meals demonstrably improved insulin resistance in subjects with T2DM and overweight or obesity. Therefore, this research could add evidence to support the idea that small changes in dietary composition can modify postprandial response, with potential clinical benefit.

Different studies have reported that isomaltulose has positive effects on glucose metabolism, decreasing glucose and insulin levels in overweight and obese subjects, as well as those with TD2M [13,20,21,22,23]. Henry et al. [24], in one randomized double-blind, controlled trial with a crossover design, including 20 men, reported that the consumption of a controlled diet combined with isomaltulose reduced the glycemic response and variability versus the combination of the same controlled diet with sucrose. In the same way, Okuno et al. [25] showed in 50 sedentary adults that the supplementation with 40 g/day of isomaltulose led to a significant reduction in insulin resistance after 12 weeks. Nevertheless, the effect of this CH in subjects with TD2M has been less studied, and the results are contradictory. For example, Brunner et al. [26] showed in T2DM subjects that the supplementation with 50 g/day of isomaltulose did not produce significant changes in HbA1c after 12 weeks of intervention. In contrast, another study including 11 participants with TD2M showed that the ingestion of 1 g/kg of isomaltulose attenuated postprandial hyperglycemia, reducing the mean plasma concentrations of insulin, C-peptide, glucagon, and glucose-dependent insulinotropic peptide by 10–23% in comparison with sucrose [27]. The mechanisms underlying the beneficial effect of isomaltulose on insulin sensitivity are not completely elucidated. Kawaguchi et al. demonstrated that the consumption of 20 g of isomaltulose improved insulin resistance in five patients with non-alcoholic fatty liver disease; they also reported that this benefit was directly associated with metabolomic changes in bile acids, fatty acids, and glycine/serine, as well as alterations in L-arginine and L-ornithine concentrations [28]. All of these factors have been previously linked to insulin signaling pathways. To our knowledge, our study is the first exploring the postprandial effects of relatively low doses of isomaltulose, and the results show that this disaccharide leads to a better postprandial glycemia than the conventional CH usually contained in beer—mainly starch and oligosaccharides, such as maltose or maltotriose. A synergistic effect of isomaltulose with the resistant dextrin should be considered when interpreting the better glycemic response after the consumption of the modified alcohol-free beer.

Different randomized controlled trials have demonstrated that resistant dextrin supplementation can modulate inflammation, improve insulin resistance, and have beneficial effects on mental health and immune system response in women with T2DM [14,29]. In this sense, Li et al. [13] demonstrated that the ingestion of 17 g of resistant dextrin led to an 18% decrease in insulin concentration in healthy subjects, which was statistically significant when compared to the control group. In the same way, another study reported a significant 22.8% decrease in insulin, 24.9% in HOMA-IR, 0.6% in glucose, and 9.6% in HbA1c concentrations in 55 women with T2DM who consumed 10 g of resistant dextrin for 8 weeks [14]. Although the mechanisms responsible for these potential benefits of this dietary fiber are not entirely known, our group has previously demonstrated that the consumption of the alcohol-free beer with almost complete fermentation of CH and the addition of isomaltulose and a resistant dextrin significantly impacted the gut microbiota in diabetic subjects with overweight or obesity, producing a significant decrease in the abundance of the genus *Parabacteroides* [30]. This bacterial genus is found in lower prevalence in patients with obesity and different metabolic diseases, and has been demonstrated to increase with the Mediterranean diet [31]. Other researchers speculate that the prebiotic supplementation favors the differentiation of L cells, which promote the secretion of digestive hormones such as GLP-1, PY, and GIP, among others [32].

GIP and GLP-1 are incretin hormones secreted from intestinal K cells and intestinal endocrine L cells, respectively, after the ingestion of a fat-rich and high-CH diet. Secretion of GLP-1 and GIP enables actions in various organs, such as the pancreas, stomach, heart, brain, and liver, among others. In the pancreas, GLP-1 and GIP have analogous actions, producing an increase in insulin biosynthesis and secretion, and decreasing apoptosis of β-cells [15]. As we hypothesized, Study 1 showed that the consumption of modified alcohol-free beers, enriched with isomaltulose and resistant dextrin, produced a lower carbohydrate overload which, in turn, produced lower peaks of insulin and incretin hormones compared to standard alcohol-free beer. In the same way, Study 2 demonstrated that the consumption of IMB with white bread produced a lower peak of glucose and, consequently, lower increases in insulin, GIP, and GLP-1 than the consumption of white bread with standard alcohol-free beer. As previously stated by other research, it seems that delaying the absorption rate of CH will influence the release of GLP-1 and GIP. These results are consistent with those of Keyhani-Nejah et al. [33], who reported in a randomized crossover study with 35 subjects (15 healthy, 10 prediabetic, and 10 T2DM) that the ingestion of 50 g of CH from isomaltulose produced significantly lower peaks of glucose and insulin, along with a lower AUC of GIP. Another randomized controlled trial included 16 subjects with TD2M, who ingested 25 g of CH coming from three different solutions: a standard solution, a; solution with resistant maltodextrin and sucromalt, and another one with lactose, isomaltulose, and resistant starch. The authors reported that the consumption of the solution containing isomaltulose produced a lower peak of GLP-1 [34]. These authors reported that a higher increase in GIP was produced after the consumption of the standard solution and the solution with isomaltulose. Nevertheless, it is important to highlight that the solution containing isomaltulose also had lactose, although the percentage of each CH contained in the solution was not reported.

None of the assessed interventions altered serum postprandial ghrelin, leptin, glucagon, pancreatic polypeptide, peptide Y, HDL cholesterol, LDL cholesterol, TG, or NEFA concentrations during the 2 h study period. These findings are not consistent with those of our previous study, in which participants reported higher satiety scores during the period of drinking the alcohol-free beer with a modified CH composition, when compared to regular alcohol-free beer [11]. The lack of a postprandial acute effect of these alcohol-free beers’ consumption on satiety hormones would not completely dismiss a mid-term effect. It is well established that the gut microbiome plays an essential role in satiety and appetite regulation [35]. If the effect of the resistant dextrin on the gut microbiome can influence satiety, regulation should be explored in the future in order to elucidate this issue. It is also important to note that in our previous study, the satiety assessment was performed using visual analog scales, which are a subjective tool. The absence of changes in lipid concentration is consistent with previously reported findings [36,37].

Our study has some limitations worth mentioning. The relatively small sample size could have limited the significance of the acute effect of each type of alcohol-free beer in some outcomes, as well as the extrapolation of findings. However, the sample size was calculated based on previous studies involving isomaltulose and resistant dextrin supplementation, with similar or even smaller sample size [34].

## 5. Conclusions

Two major conclusions could be extrapolated from this study: Firstly, the consumption of alcohol-free beers with almost complete fermentation of regular CH, enriched with isomaltulose (2.5 g/100 mL) and resistant maltodextrin (0.8 g/100 mL), or enriched exclusively with resistant maltodextrin (2.0 g/100 mL), produced a lower peak of glucose, insulin, and incretin hormones than the consumption of regular alcohol-free beer. Secondly, the consumption of the modified alcohol-free beer enriched with isomaltulose and resistant dextrin, together with white bread, generates a lower glucose peak than the consumption of the same quantity of CH when sourced entirely from white bread. This demonstrates that this modified alcohol-free beer could sustainably influence and improve the glycemic postprandial effect of a meal. This could lead to clinical benefits in terms of glycemic metabolism in subjects with T2DM, as previously demonstrated by our group, by increasing the evidence that small changes in nutritional composition are a cornerstone when improving the management or prevention of glycemic disorders.

## Figures and Tables

**Figure 1 nutrients-14-01046-f001:**
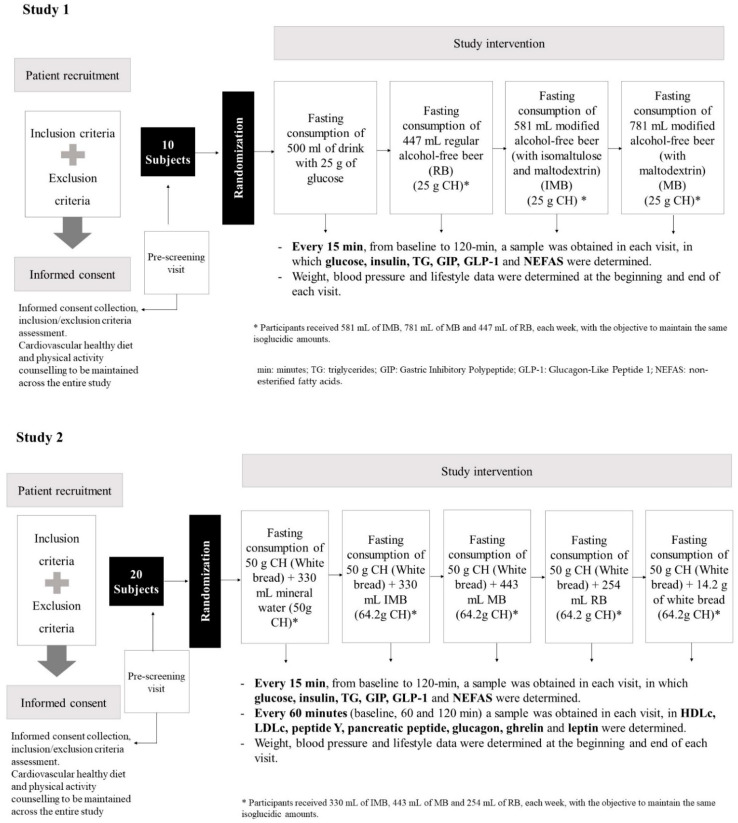
Study design description.

**Figure 2 nutrients-14-01046-f002:**
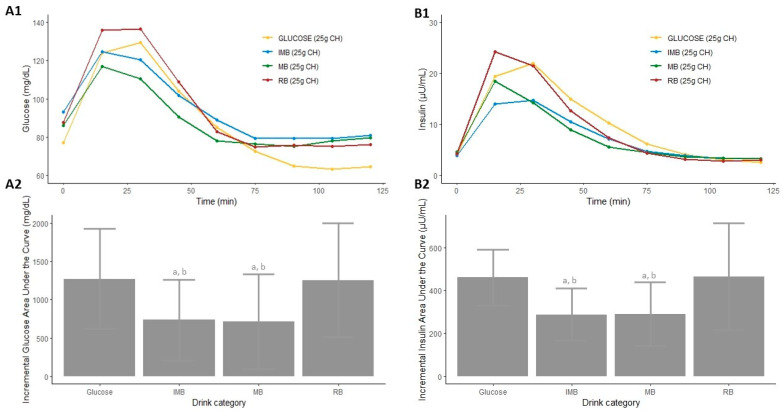
Serum postprandial glucose concentrations (**A1**), insulin concentrations (**B1**), incremental glucose AUC (**A2**), and incremental insulin AUC (**B2**) after the consumption of each type of drink in Study 1. Differences in the incremental glucose and insulin AUC (**A2**,**B2**) after the consumption of each type of drink in Study 1 were determined with the use of a mixed-design ANOVA model. The Tukey–Kramer method was used for the post hoc analyses. a: *p*-value < 0.05 comparing the iAUC of IMB or MB vs. glucose. b: *p*-value < 0.05 comparing the iAUC of IMB or MB vs. RB. CH: carbohydrates; IMB: alcohol-free beer with almost complete fermentation of regular CH, enriched with isomaltulose (2.5 g/100 mL) and a resistant maltodextrin (0.8 g/100 mL); MB: alcohol-free beer with almost complete fermentation of regular CH, enriched with maltodextrin (2.0 g/100 mL); RB: regular alcohol-free beer.

**Figure 3 nutrients-14-01046-f003:**
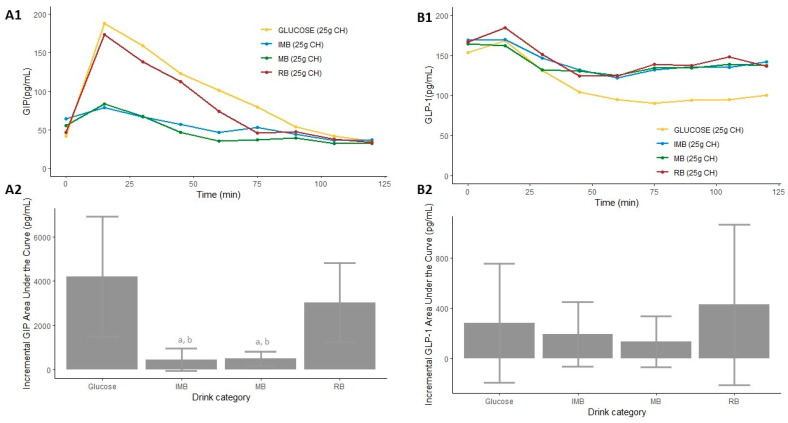
Serum postprandial GIP concentrations (**A1**), GLP-1 concentrations (**B1**), incremental GIP AUC (**A2**), and incremental GLP-1 AUC (**B2**) after the consumption of each type of drink in Study 1. Differences in the incremental GIP and GLP-1 AUC (**A2**,**B2**) after the consumption of each type of drink in Study 1 were determined with the use of a mixed-design ANOVA model. The Tukey–Kramer method was used for the post hoc analyses. a: *p*-value < 0.05 comparing the iAUC of IMB or MB vs. glucose. b: *p*-value < 0.05 comparing the iAUC of IMB or MB vs. RB. CH: carbohydrates; IMB: alcohol-free beer with almost complete fermentation of regular CH, enriched with isomaltulose (2.5 g/100 mL) and a resistant maltodextrin (0.8 g/100 mL); MB: alcohol-free beer with almost complete fermentation of regular CH, enriched with maltodextrin (2.0 g/100 mL); RB: regular alcohol-free beer; GIP: gastric inhibitory polypeptide; GLP-1: glucagon-like peptide 1.

**Figure 4 nutrients-14-01046-f004:**
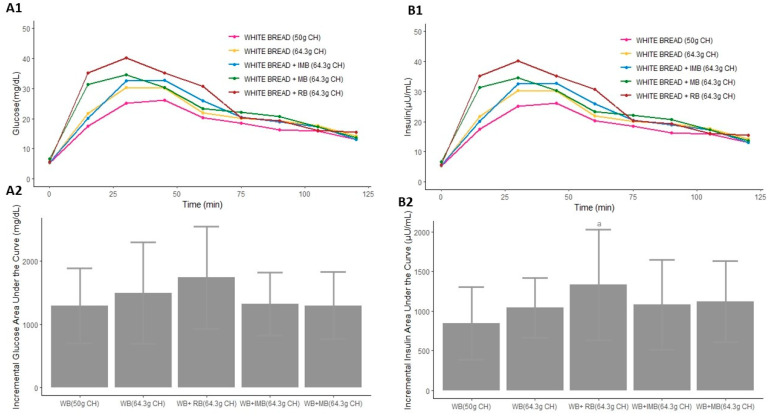
Serum postprandial glucose concentrations (**A1**), insulin concentrations (**B1**), incremental glucose AUC (**A2**), and incremental insulin AUC (**B2**) after the consumption of each type of drink combined with white bread in Study 2. Differences in the incremental glucose and insulin AUC (**A2**,**B2**) after the consumption of each type of drink in Study 1 were determined with the use of a mixed-design ANOVA model. The Tukey–Kramer method was used for the post hoc analyses. a: *p*-value < 0.05 comparing the iAUC of WB + RB (64.3 g CH) vs. WB (50 g CH). CH: carbohydrates; WB: white bread; IMB: alcohol-free beer with almost complete fermentation of regular CH, enriched with isomaltulose (2.5 g/100 mL) and a resistant maltodextrin (0.8 g/100 mL); MB: alcohol-free beer with almost complete fermentation of regular CH, enriched with maltodextrin (2.0 g/100 mL); RB: regular alcohol-free beer.

**Figure 5 nutrients-14-01046-f005:**
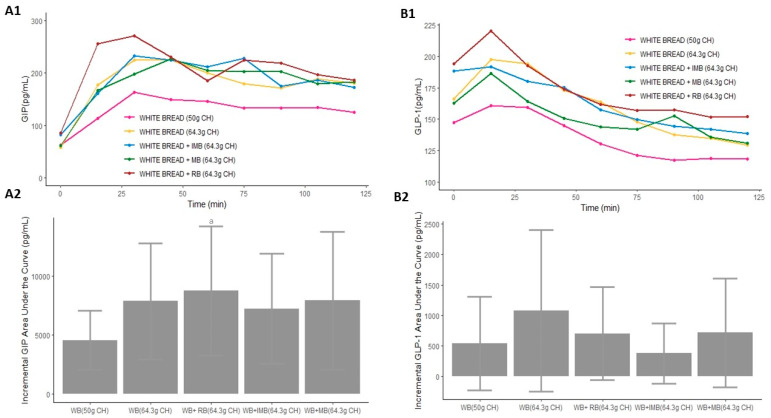
Serum postprandial GIP concentrations (**A1**), GLP-1 concentrations (**B1**), incremental GIP AUC (**A2**), and incremental GLP-1 AUC (**B2**) after the consumption of each type of drink combined with white bread in Study 2. Differences in the incremental GIP and GLP-1 AUC (**A2**,**B2**) after the consumption of each type of drink in Study 1,were determined with the use of a mixed-design ANOVA model. The Tukey–Kramer method was used for the post hoc analyses. a: *p*-value < 0.05 comparing the iAUC of RB + white bread (64.3 g CH) vs. white bread (50 g CH). CH: carbohydrates; IMB: alcohol-free beer with almost complete fermentation of regular CH, enriched with isomaltulose (2.5 g/100 mL) and a resistant maltodextrin (0.8 g/100 mL); MB: alcohol-free beer with almost complete fermentation of regular CH, enriched with maltodextrin (2.0 g/100 mL); RB: regular alcohol-free beer; GIP: gastric inhibitory polypeptide; GLP-1: glucagon-like peptide 1.

**Figure 6 nutrients-14-01046-f006:**
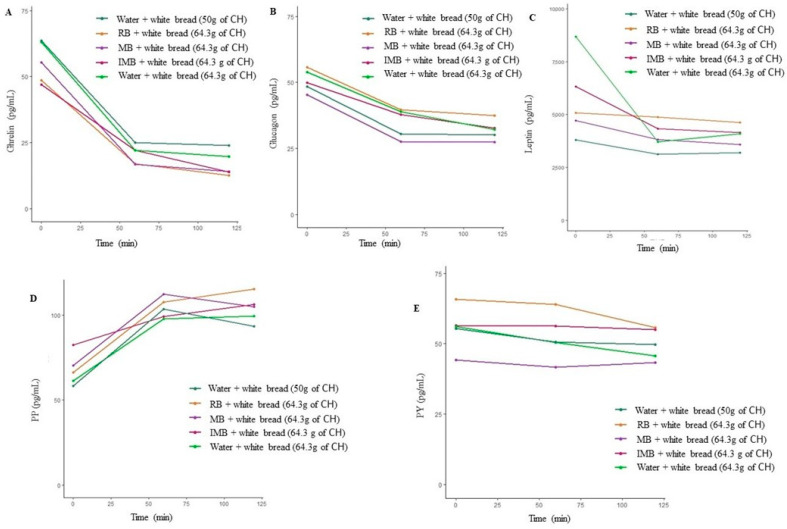
Ghrelin (**A**), glucagon (**B**), leptin (**C**), PP (**D**), and PY (**E**) over time, depending on the type of drink consumed in Study 2. CH: carbohydrates; IMB: alcohol-free beer with almost complete fermentation of regular CH, enriched with isomaltulose (2.5 g/100 mL) and a resistant maltodextrin (0.8 g/100 mL); MB: alcohol-free beer with almost complete fermentation of regular CH, enriched with maltodextrin (2.0 g/100 mL); RB: regular alcohol-free beer; PY: peptide Y; PP: pancreatic polypeptide.

**Table 1 nutrients-14-01046-t001:** Clinical, anthropometric, and biochemical characteristics of all subjects included in both studies.

	Subjects Included in Study 1 (*N* = 10)	Subjects Included in Study 2 (*N* = 20)
Age, years	31.5 (30.3–34.5)	30.5 (24.8–33.0)
Men, *n* (%)	7 (70%)	14 (70%)
Body weight, kg	86.0 (80.8–87.8)	76.7 (61.4–84.7)
BMI, kg/m^2^	23.4 (22.4–25.7)	24.4 (20.6–26.0)
Alcohol consumption (g/d)	2.35 (0.47–5.81)	6.25 (1.22–15.2)
TSH, μUI/L	2.11 ± 0.99	2.36 ± 1.28
Triglycerides, mg/dL	56.0 (51.0–63.3)	76.0 (47.5–89.0)
Total cholesterol, mg/dL	192 (183–198)	188 (168–199)
LDL cholesterol, mg/dL	113 (103–122)	112 (98.0–128)
HDL cholesterol, mg/dL	61.0 (53.5–69.3)	56.0 (50.5–60.8)
GGT, UI/L	13.0 (13.0–16.3)	18.5 (15.5–23.0)
GOT, UI/L	22.0 (21.0–25.5)	25.0 (21.0–28.3)
GPT, UI/L	18.0 (14.0–22.3)	17.5 (14.0–29.3)
Glucose, mg/dL	86.0 (80.8–87.8)	86.0 (82.3–95.0)

Quantitative variables are expressed as means ± standard deviations, except for variables not following normal distribution, which are expressed as medians (25th percentile–75th percentile). Qualitative variables are expressed as percentages. BMI: body mass index; GGT: gamma glutamyl transferase; GOT: glutamic oxaloacetic transaminase; GPT: glutamic–pyruvic transaminase; TSH: thyroid-stimulating hormone.

**Table 2 nutrients-14-01046-t002:** Maximum increase of glucose, insulin, GIP, and GLP-1 produced after the consumption of each drink/meal in both studies.

	Study 1 (*N* = 10)					Study 2 (*N* = 20)					
	Regular Alcohol-Free Beer (RB) (25 g of CH)	Alcohol-Free Beer with Isomaltulose and Resistant Maltodextrin (IMB) (25 g of CH)	Alcohol-Free Beer with Resistant Maltodextrin (MB) (25 g of CH)	Glucose Solution (25 g of CH)	*p*	RB + 50 g of CH from White Bread (64.3 g of CH)	IMB + 50 g of CH from White Bread (64.3 g of CH)	MB + 50 g of CH from White Bread (64.3 g of CH)	Water+ 50 g of CH from White Bread (50 g of CH)	Water + 64.3 g of CH from White Bread (64.3 g of CH)	*p*
Glucose variation, mg/dL	60.4 ± 24.1	33.7 ± 14.2 *	34.7 ± 19.1 *	58.0 ± 20.0	0.005	58.3 ± 14.9	44.7 ± 9.83 *	41.1 ± 10.8 *	41.2 ± 13.2	48.1 ± 14.8	0.002
Insulin variation, mg/dL	22.2 ± 12.2	12.1 ± 4.31 *	14.8 ± 8.6 **^+^**	18.8 ± 7.48	0.012	40.1 ± 17.8	31.1 ± 11.7 ^+^	35.9 ± 19.4	27.7 ± 14.9	30.5 ± 11.4	0.124
GIP variation, pg/mL	128 ± 62.3	18.9 ± 23.8 *	34.6 ± 22.1 *	168 ± 112	<0.001	229 ± 134	240 ± 189	229 ± 194	126 ± 41.4	216 ± 185	0.011
GLP-1 variation, pg/mL	17.9 ± 23.7	3.27 ± 33.0	1.67 ± 33.7	13.9 ± 35.0	0.529	27.0 ± 47.3	13.3 ± 33.7	28.8 ± 29.0	21.3 ± 30.9	46.6 ± 45.1	0.058

Maximum increase in each analyte was calculated as the difference between the maximum value reached over 120 min and the baseline value. Quantitative variables were expressed as the mean ± standard deviation. CH: carbohydrates; IMB: alcohol-free beer with almost complete fermentation of regular CH, enriched with isomaltulose (2.5 g/100 mL) and a resistant maltodextrin (0.8 g/100 mL); MB: alcohol-free beer with almost complete fermentation of regular CH, enriched with maltodextrin (2.0 g/100 mL); RB: regular alcohol-free beer.^.^ The *p*-value was calculated by the Kruskal–Wallis test. * *p* < 0.05; ^+^
*p* < 0.090, comparing IMB vs. RB or MB vs. RB, as applicable, calculated by *t*-test.

**Table 3 nutrients-14-01046-t003:** Maximum increases in ghrelin, glucagon, leptin, PP, and PY produced after the consumption of each drink/meal in Study 2.

**GHRELIN**	**Baseline**	**60 min**	**120 min**	***p* ^1^**
RB + white bread (64.3 g of CH)	49.3 (16.2–64.3)	11.8 (10.4–18.4)	11.6 (10.6–13.0)	0.942
IMB + white bread (64.3 g of CH)	48.8 (31.3–84.3)	13.0 (12.1–17.3)	13.0 (10.6–13.8)	
MB + white bread (64.3 g of CH)	36.6 (23.9–65.3)	13.7 (11.8–27.3)	13.0 (11.5–16.4)	
Water + white bread (64.3 g of CH)	57.8 (19.2–90.3)	12.6 (10.6–17.9)	13.0 (10.6–17.2)	
Water+ white bread (50 g of CH)	51.4 (20.6–74.7)	17.9 (12.9–34.0)	13.0 (11.0–33.3)	
**GLUCAGON**	**Baseline**	**60 min**	**120 min**	***p* ^1^**
RB + white bread (64.3 g of CH)	53.0 (44.1–60.0)	37.5 (24.4–53.2)	33.4 (19.6–47.2)	0.988
IMB + white bread (64.3 g of CH)	43.9 (36.1–57.8)	23.0 (16.8–35.7)	24.9 (15.0–37.4)	
MB + white bread (64.3 g of CH)	47.6 (38.2–62.5)	34.0 (27.1–47.5)	33.6 (18.1–40.3)	
Water + white bread (64.3 g of CH)	46.9 (36.9–61.5)	31.5 (23.4–53.4)	30.3 (20.6–39.2)	
Water+ white bread (50 g of CH)	45.9 (36.4–54.5)	28.4 (19.1–42.9)	29.7 (15.0–39.5)	
**LEPTIN**	**Baseline**	**60 min**	**120 min**	***p* ^1^**
RB + white bread (64.3 g of CH)	3684 (1386–6801)	2714 (1243–6057)	3079 (1145–5408)	0.723
IMB + white bread (64.3 g of CH)	2732 (980–5038)	2195 (824–4047)	2066 (819–3804)	
MB + white bread (64.3 g of CH)	3101 (1150–6119)	2450 (1178–4567)	2414 (1056–4389)	
Water + white bread (64.3 g of CH)	2576 (1372–7047)	2030 (1087–5380)	2281 (1033–6204)	
Water+ white bread (50 g of CH)	2030 (1374–4289)	1977 (1177–3609)	1861 (1056–3533)	
**PP**	**Baseline**	**60 min**	**120 min**	***p* ^1^**
RB + white bread (64.3 g of CH)	47.1 (31.4–65.7)	79.9 (65.7–105)	83.8 (57.4–132)	0.982
IMB + white bread (64.3 g of CH)	47.2 (23.5–56.7)	82.0 (52.8–122)	71.0 (39.2–120)	
MB + white bread (64.3 g of CH)	49.5 (37.1–88.6)	76.8 (46.3–127)	94.7 (47.4–114)	
Water + white bread (64.3 g of CH)	43.8 (35.5–58.7)	70.2 (47.8–110)	78.3 (63.4–112)	
Water+ white bread (50 g of CH)	36.0 (30.2–55.3)	84.3 (41.7–117)	65.6 (38.4–99.5)	
**PY**	**Baseline**	**60 min**	**120 min**	***p* ^1^**
RB + white bread (64.3 g of CH)	13.0 (13.0–93.2)	30.0 (13.0–85.5)	13.0 (13.0–78.9)	0.992
IMB + white bread (64.3 g of CH)	13.0 (13.0–58.9)	13.0 (13.0–51.1)	13.0 (13.0–58.4)	
MB + white bread (64.3 g of CH)	13.0 (13.0–97.2)	13.0 (13.0–106)	13.0 (13.0–101)	
Water + white bread (64.3 g of CH)	13.0 (13.0–97.8)	20.5 (13.0–69.8)	13.0 (13.0–62.5)	
Water+ white bread (50 g of CH)	26.8 (13.0–99.3)	13.0 (13.0–102)	13.0 (13.0–88.3)	

The maximum increase in each analyte was calculated as the difference between the maximum value reached over 120 min and the baseline value. Quantitative variables are expressed as medians (25th percentile–75th percentile). CH: carbohydrates; IMB: alcohol-free beer with almost complete fermentation of regular CH, enriched with isomaltulose (2.5 g/100 mL) and a resistant maltodextrin (0.8 g/100 mL); MB: alcohol-free beer with almost complete fermentation of regular CH, enriched with maltodextrin (2.0 g/100 mL); RB: regular alcohol-free beer; PY: peptide Y; PP: pancreatic polypeptide. ^1^ The *p*-value was calculated using the Kruskal–Wallis test.

## Data Availability

The database generated as a result of the systematic review carried out here will be fully available to replicate the results that are necessary at the request of the reviewers or editors.

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
