# Peer review of "Effect of the Consumption of Alcohol-Free Beers with Different Carbohydrate Composition on Postprandial Metabolic Response"

_nutrients, 2022, doi:10.3390/nu14051046_

Round 1

Reviewer 1 Report

Manuscript titled “Effect of the consumption of alcohol-free beers with different carbohydrate composition on postprandial metabolic response” by Lamiquiz-Moneo et al. investigated the postprandial effects of regular alcohol-free beer to alcohol-free beer (RB) with modified carbohydrates (CH) composition using two different studies in healthy volunteers. A detailed analysis on of many blood metabolites, including glucose, lipids, insulin, GLP, GIP were monitored at different timepoints drinking RB or modified-CH.

The subject being investigated is of significance with the current number of patients affected by T2DM and related comorbidities. 

In fact, this paper is very much in line, a follow up but on healthy subjects, with previous papers published by the same group on the effects of adding modified carbohydrates (CH) to alcohol free beer to insulin resistant/diabetic patients and the gut microbiome of these patients.

A few minor comments to manuscript:

1) Even if the sample size is limited, one of the key points of this study is the unique contribution of resistant maltodextrin and isomaltose in improving glycemic index and insulin compared to the glucose-based drink and RB. However, in study 2, no significant changes were detected in many parameters measured when comparing RB to modified CH beer together with white bread.

A) Would increasing the amount of isomaltose +/- resistant maltodextrin improve the AUC for glucose and insulin with bread consumption?

B) How was the dose for isomaltose +/- resistant maltodextrin chosen? In the published literature as pointed out by the authors, higher amounts of both compounds have been used (lines 56-75,page 3 of 18).

C) Are there any known toxic effects at high (what is high?) concentrations of isomaltose and/or resistant maltodextrin.

2) On line 39-40 of page 3 of 18, the authors draw conclusion that doubling the dose of resistant maltodextrin (MB), but not isomaltose (IMB) does lower glucose and insulin peak. However, no data in this manuscript shows any effects of doubling the dose of isomaltose alone and compared to resistant maltodextrin.

Altogether, the paper is very nicely written and overall, all the data were of high quality. Many aspects of the study are of descriptive nature. Although the authors allude to a possible mechanism to why the modified CH could be beneficial and healthier (line 76, page 4 of 18), further investigation could be done to provide mechanistic depth. Perhaps with the blood samplings collected in both studies, a blood-based gene expression that could highlight signaling pathways would be very impactful.

Cheers!

Author Response

Reviewer 1

Manuscript titled “Effect of the consumption of alcohol-free beers with different carbohydrate composition on postprandial metabolic response” by Lamiquiz-Moneo et al. investigated the postprandial effects of regular alcohol-free beer to alcohol-free beer (RB) with modified carbohydrates (CH) composition using two different studies in healthy volunteers. A detailed analysis on of many blood metabolites, including glucose, lipids, insulin, GLP, GIP were monitored at different timepoints drinking RB or modified-CH.

The subject being investigated is of significance with the current number of patients affected by T2DM and related comorbidities. 

In fact, this paper is very much in line, a follow up but on healthy subjects, with previous papers published by the same group on the effects of adding modified carbohydrates (CH) to alcohol free beer to insulin resistant/diabetic patients and the gut microbiome of these patients.

A few minor comments to manuscript:

  1. A) Even if the sample size is limited, one of the key points of this study is the unique contribution of resistant maltodextrin and isomaltose in improving glycaemic index and insulin compared to the glucose-based drink and RB. However, in study 2, no significant changes were detected in many parameters measured when comparing RB to modified CH beer together with white bread.

We agree with the reviewer that the sample size is limited. Albeit, it was calculated with an IC 95% using the data reported in the first study and this one was estimated according to previous research that also involved small samples trials.

On the other hand, we found in the first study that the consumption of RB produced higher AUC of glucose, insulin and GIP than the AUC produced after the consumption to modified CH beer. Likewise, we can see similar effects in the second study. For example, the AUCs of glucose after the consumption of modified CH beer with white bread were significantly lower than that produced after the consumption of RB plus white bread. However, it is true that we did not find significantly differences in insulin or GIP. That could be explained because the consumption of each beer is lower than in the first study, which would indicate that the effect of modified CH beer depends on the amount of isomaltulose and maltodextrin intake. Nevertheless, it is really interesting to highlight that the consumption of modified CH beer plus white bread produced significantly lower AUC that the consumption of the same amount of CH, all of them coming from white bread. This finding would demonstrate that isomaltulose and resistant maltodextrin (even in lower doses) could modulate postprandial response of a meal.

  1. B) Would increasing the amount of isomaltose +/- resistant maltodextrin improve the AUC for glucose and insulin with bread consumption?

Yes, we consider that increasing the amount of isomaltulose and maltodextrin would improve the AUC of glucose and insulin with bread consumption. However, as we have mentioned before, with that dose of isomaltulose in the modified CH beer ingested with white bread lead to lower AUC than the produced after the consumption of the same CH, coming all of them from white bread. It is important to note that isomaltulose is a disaccharide providing 4 kcal/g and an over intake could counteract these benefits mainly in subjects with prediabetes and type 2 diabetes. In the same way, higher doses of resistant maltodextrin could produce gastrointestinal discomfort as flatulence or diarrhea, as previously reported by other trials. Thus, although we agree with the reviewer that higher amount of isomaltulose and maltodextrin could generate even higher benefits in the glycaemic control, we aimed to explore if low doses of these compounds would reach significant postprandial effect by optimizing clinical benefits.

  1. C) How was the dose for isomaltose +/- resistant maltodextrin chosen? In the published literature as pointed out by the authors, higher amounts of both compounds have been used (lines 56-75,page 3 of 18).

Isomaltulose is a functional carbohydrate composed of glucose and fructose with the same caloric value as sucrose of 4kcal/g. However, the digestive rate of isomaltulose is four to five times slower than that of sucrose because of the strong glucose–fructose a-1,6-glycosidic bond (Emilia SokoÅ‚owska et al. Crit Rev Food Sci Nutr. 2021). Different studies, previously published, have demonstrated a significant effect by adding higher doses of isomaltulose, for example the addition of 40g/day in 15 subjects generated higher decreased of fat mass percentage than the consumption of 40g/day of sucrose (Lightowler et al. Nutrients 2019. PMID:31590285). As previously stated, we aimed to explore if lower doses of these compounds could reach clinical benefits by optimizing their intake.

  1. D) Are there any known toxic effects at high (what is high?) concentrations of isomaltose and/or resistant maltodextrin.

There are not describe toxic effects of high doses of isomaltulose. However, it is pivotal to highlight that the isomaltulose is a sugar, and as such, its consumption in high doses entails the intake of a relatively high number of calories. For this, the addition of high doses of isomaltulose in these modified CH beers generated a high calorie beer, which would be completely discouraged in diabetic patients. The study developed by Ang et al. (Am J Clin Nutr. 2014 Oct;100(4):1059-68.) involved an intake of 1 g/kg of body weight of isomaltulose and no adverse events were reported.

Regarding to resistant maltodextrin, although other studies have shown that higher dose of resistant maltodextrin could produce diarrhea, these doses should be really elevated, higher than 0,8-1g of maltodextrin per kg of body weight (Yoshikawa et al. J Toxicol Sci 2013). For this reason, the maltodextrin resistant doses were chosen with the idea of preserving the original value of the beer and try to find the minimal dose of this fibre which could be generate positive effects, with the idea that increase the amount of maltodextrin could generate intestinal problems, such as diarrhea or abdominal distension.

  1. E) On line 39-40 of page 3 of 18, the authors draw conclusion that doubling the dose of resistant maltodextrin (MB), but not isomaltose (IMB) does lower glucose and insulin peak. However, no data in this manuscript shows any effects of doubling the dose of isomaltose alone and compared to resistant maltodextrin.

We are really sorry, but we are not sure if we completely understand the comment and we have not been able to find this conclusion in the page 3 of 18. We guess that you referred to page 13 where the main conclusion of the study is included.

One of the conclusions of this study was that the modified CH alcohol-free beer led to lower AUC of glucose than the glucose-based beverage. This effect was observed both in the alcohol-free beer containing isomaltulose and 0.8 g/100 mL of resistant maltodextrin and in the alcohol-free beer with 2 g/100 mL of resistant maltodextrin but no isomaltulose. For that reason, we referred to the double dose of this compound that reached the same postprandial benefit that the beer containing both compounds.

We did not explore the effect of a modified alcohol-free beer with the doble dose of isomaltulose because, as previously stated, this a disaccharide providing 4 kcal/g. Thus, higher dose could lead to greater benefit but we hypnotized that the over intake of calories could counteract the potential benefit.

Altogether, the paper is very nicely written and overall, all the data were of high quality. Many aspects of the study are of descriptive nature. Although the authors allude to a possible mechanism to why the modified CH could be beneficial and healthier (line 76, page 4 of 18), further investigation could be done to provide mechanistic depth. Perhaps with the blood samplings collected in both studies, a blood-based gene expression that could highlight signalling pathways would be very impactful.

We appreciate the useful comments of the reviewer. We agree that we could investigate the determination of blood-based gene expression for studying possible signalling pathways. However, we would like to highlight that this article is the third study that we have developed with this modified HC beers. First and foremost, we demonstrated that the regular consumption of this modified HC beer combined with healthy diet generated greater reduction of body weight and glucose levels than the consumption of the same healthy diet with regular beer. Further this, we showed that the regular consumption of this type of beer produced significant changes in the gut microbiome. 

Reviewer 2 Report

Please see the attached file for my comments. The idea and concept is novel.
